# Augmenting the Angiogenic Profile and Functionality of Cord Blood Endothelial Colony-Forming Cells by Indirect Priming with Bone-Marrow-Derived Mesenchymal Stromal Cells

**DOI:** 10.3390/biomedicines11051372

**Published:** 2023-05-05

**Authors:** Ashutosh Bansal, Archna Singh, Tapas Chandra Nag, Devyani Sharma, Bhavuk Garg, Neerja Bhatla, Saumitra Dey Choudhury, Lakshmy Ramakrishnan

**Affiliations:** 1Department of Cardiac Biochemistry, All India Institute of Medical Sciences, New Delhi 110029, India; ashutosh.bansal0408@gmail.com; 2Department of Biochemistry, All India Institute of Medical Sciences, New Delhi 110029, India; 3Department of Anatomy, All India Institute of Medical Sciences, New Delhi 110029, India; 4Department of Orthopaedics, All India Institute of Medical Sciences, New Delhi 110029, India; 5Department of Obstetrics & Gynaecology, All India Institute of Medical Sciences, New Delhi 110029, India; 6Centralized Core Research Facility, All India Institute of Medical Sciences, New Delhi 110029, India

**Keywords:** endothelial colony-forming cells (ECFCs), bone-marrow-derived mesenchymal stem/stromal cells (BM-MSCs), co-culture, direct priming, indirect priming, functionality of ECFCs, proteome profiling

## Abstract

Cellular therapy has shown promise as a strategy for the functional restoration of ischemic tissues through promoting vasculogenesis. Therapy with endothelial progenitor cells (EPCs) has shown encouraging results in preclinical studies, but the limited engraftment, inefficient migration, and poor survival of patrolling endothelial progenitor cells at the injured site hinder its clinical utilization. These limitations can, to some extent, be overcome by co-culturing EPCs with mesenchymal stem cells (MSCs). Studies on the improvement in functional capacity of late EPCs, also referred to as endothelial colony-forming cells (ECFCs), when cultured with MSCs have mostly focused on the angiogenic potential, although migration, adhesion, and proliferation potential also determine effective physiological vasculogenesis. Alteration in angiogenic proteins with co-culturing has also not been studied. We co-cultured ECFCs with MSCs via both direct and indirect means, and studied the impact of the resultant contact-mediated and paracrine-mediated impact of MSCs over ECFCs, respectively, on the functional aspects and the angiogenic protein signature of ECFCs. Both directly and indirectly primed ECFCs significantly restored the adhesion and vasculogenic potential of impaired ECFCs, whereas indirectly primed ECFCs showed better proliferation and migratory potential than directly primed ECFCs. Additionally, indirectly primed ECFCs, in their angiogenesis proteomic signature, showed alleviated inflammation, along with the balanced expression of various growth factors and regulators of angiogenesis.

## 1. Introduction

Cardiovascular diseases (CVDs), predominantly ischemic heart disease (IHD) and stroke, rank first among the causes of mortalities in developing countries [1,2,3], with India experiencing an alarming rise in deaths due to an increase in heart attacks of over 53% in just 5 years [4]. To alleviate the ischemic burden, a promising therapeutic approach is to restore blood flow to the infarcted tissue. This may either be achieved by stabilizing the endothelium present in the lesion zone, which is generally difficult to achieve, or by mobilizing endothelial progenitor cells (EPCs) to the ischemic tissue [5]. The capability of EPCs to self-renew, and their direct participation in in vivo vasculogenesis [6,7,8], has gained the attention of researchers globally. EPCs in circulation are known to exist as two subsets: (i) early EPCs, having a hematopoietic origin and sharing a genetic profile similar to monocytes, mostly involved in angiogenesis through paracrine mechanisms, and can never give rise to mature endothelial cells [9,10,11,12]; (ii) late EPCs, or endothelial colony-forming cells (ECFCs), which display a genetic signature similar to endothelial cells, and are shown to be actively involved in tube formation [9,13]. Despite the promising results of EPCs in neovascularization in preclinical studies [13,14,15,16,17,18,19,20], the use of EPCs in clinical trials has shown less promise due to their low frequency of occurrence, lengthy culture time required to achieve a sufficient therapeutic dose, limited survival following administration, and the requirement of autologous administration due to their inherent immunogenicity [21]. To add to this, the number and function of ECFCs in circulation deteriorate significantly in disease conditions [22,23,24,25], further limiting their autologous usage.

To overcome some of these problems, dual cell therapy has been suggested. During developmental stages, ECFCs reside in a common niche with mesenchymal stromal cells (MSCs) in bone marrow [26]. In adulthood, endothelial cells in capillaries maintain a close association with pericytes, which exhibit many similarities with MSCs [27,28], and any impairment in MSCs has been linked to the onset of various vascular disorders [27,29]. This suggests that MSCs may play a stabilizing role over ECFCs, highlighting the importance of understanding the cellular-level mechanisms involved in this interaction. Co-culturing ECFCs with MSCs has been shown to provide several benefits, including reduced immunogenic response elicited by ECFC post-allogeneic transplantation and increased survival [30], rapidity in blood perfusion in vivo [14,31], and an increase in their vasculogenic potential, independent of the origin of the MSCs [32]. While experimental vasculogenesis is important, it alone is not sufficient to fully capture the biological process of angiogenesis, which involves the recruitment, survival, and proliferation of circulating progenitor cells at the site of injury, their migration towards the ischemic lesion, and the formation of functional capillaries in the ischemic zone. Therefore, to fully understand the benefits of co-culturing ECFCs with MSCs, we conducted in vitro experiments under conditions that mimic the physiological milieu during cardiovascular disease, and evaluated various functional aspects of ECFCs, including adhesion potential, migratory ability, proliferative potential, and vasculogenic ability. Furthermore, we also quantified the expression of different proteins implicated in each of the above-stated processes, including the profile of various pro- and anti-angiogenic factors and inflammation. We aimed to obtain a more comprehensive understanding of the biological signaling that mediates the influence of MSCs on ECFCs at each stage of the angiogenesis process, which is essential for devising cellular modulation strategies for regenerative medicine and tissue engineering.

## 2. Materials and Methods

### 2.1. Ethics Statement 

Ethical approvals were obtained from the Institutional Ethics Committee (IEC) with Ref. No.: IEC-177/11.04.2020, RP-20/2020, RP-17/2020 dated 15 May 2020 and the Institutional Committee for Stem Cell Research (ICSCR) with Ref No.: IC-SCR/105/20(o) dated 2 June 2020, All India Institute of Medical Science, New Delhi. All participants were informed about study procedures, and written consent was obtained before the collection of any sample used for experimental purposes. Human umbilical cord blood was collected from healthy normotensive mothers undergoing cesarean delivery. Subjects who had any pathological conditions or twins were excluded from the study. Bone marrow aspirates were collected from healthy subjects aged 22 to 30 years undergoing routine medical surgeries in the Department of Orthopaedics, All India Institute of Medical Sciences, New Delhi.

### 2.2. Isolation of Cord-Blood-Derived Endothelial Colony-Forming Cells (ECFCs)

The cord blood mononuclear cells (CBMNCs) were isolated from 10–15 mL of heparinized human umbilical cord blood samples by density gradient centrifugation using Lymphoprep, density 1.077 (04-03-9391/02, Alere Technologies, Oslo, Norway), and resuspended in endothelial basal media supplemented with endothelial growth media (EGM), SingleQuots supplements (CC-3162, Lonza, Walkersville, MD USA), and 10% fetal bovine serum (FBS) (10100147; Gibco, Australia). The mononuclear cells were seeded onto rat tail collagen type 1 (354236, Corning, Bedford, MA USA)-coated 6-well tissue culture plates (5 µg/cm^2^) at a seeding density of 1 × 10^7^ cells/well, and cultured at 37 °C and 5% CO_2_. After 5 days, non-adherent cells were aspirated, adherent cells were further cultured, and the media was changed every other day. The ECFC colonies with a cobblestone-shaped morphology appeared between 7 and 14 days of culture. For all the experiments, ECFCs at passages 3–5 were used.

### 2.3. Characterization of ECFCs

#### 2.3.1. Immunophenotyping

ECFCs, collected at passage 2, were characterized using flow cytometry as per the guidelines stated by Medina et al., 2017 [9]. The expression of endothelial antigens CD31, KDR, and CD146, progenitor antigens CD34, CD105, and CD117, and the leukocyte antigen CD45 were assessed. Briefly, the adherent ECFCs were harvested by treatment with TrypLE express enzyme (Cat. No. 12605010; Gibco), counted, resuspended in 100 µL staining buffer, and then stained with FITC-conjugated mouse anti-human CD31 (Cat. No. 303103, 1:100), FITC-conjugated anti-human CD146 (Cat. No. 361011, 1:100), PE-conjugated mouse anti-human KDR (Cat. No. 393003, 1:100), Alexa Fluor 488-conjugated mouse anti-human CD105 (Cat. 323209, 1:100), PE-conjugated CD117 (Cat. No. 340529, 1:100), APC-conjugated mouse anti-human CD34 (Cat. No. 343509, 1:100), and PerCP mouse anti-human CD45 (Cat. No. 368506, 1:100). Cells were incubated with fluorophore-conjugated antibodies at 4 °C for 30 min, and stained cells were detected with LSR Fortessa X-20 (Becton Dickinson, Franklin Lakes, NJ, USA). Cultured ECFCs were considered negative for the leukocyte marker CD45, positive for the stem cell markers CD117 and CD34, and positive for endothelial markers CD31, KDR, and CD146, respectively. 

#### 2.3.2. Immunocytochemistry Characterization

Cultured ECFCs were also assessed for KDR and CD34 expression. Briefly, cells at passage 2 were grown on 12 mm coverslips with approximately 60–70% confluency and were fixed with 4% paraformaldehyde (PFA) solution and blocked with 1% BSA, 22.52 mg/mL glycine in PBST (PBS+ 0.1% Tween 20). Cells were then incubated with mouse monoclonal anti-human KDR primary antibody (20 µg/mL) (Cat. MAB3571; R&D Systems) for 1 h, followed by goat anti-mouse secondary antibody, Alexa Fluor™ Plus 555 (10 µg/mL) (Cat. No. A32727, Invitrogen, Waltham, MA, USA) in the dark for 1 h. Thereafter, cells were incubated with Alexa Fluor 488-conjugated mouse anti-human CD34 antibody (10 µg/mL) (Cat. FAB7227G; R&D Systems, Minneapolis, MN, USA) for 1 h in the dark. The coverslips were then mounted using an aqueous mounting medium containing DAPI as the counterstain (ab104139, Abcam, Waltham, MA USA), and images were obtained using a Zeiss LSM 980 confocal microscope (Zen Blue, version 3.1). 

Additionally, the ability of cells to bind to Ulex-lectin and uptake acetylated low-density lipoprotein, Ac-LDL, was assessed. In brief, ECFCs at passage 2 were incubated with 1,1′-dioctadecyl-3,3,3′,3′-tetramethylindo-carbocyanine-labeled Ac-LDL (Dil-Acy-LDL; 5 µg/mL) (Cat. L3484; Life Technologies, Grand Island, NY, USA) and fixed with 4% PFA. These cells were then incubated with FITC-labeled Ulex europaeus agglutinin (FITC-UEA-1; 10 µg/mL) (Cat. FL-1061; Vector Laboratories, Newark, CA, USA). After rinsing with PBS, the coverslips were mounted using an aqueous mounting medium containing DAPI as the counterstain (ab104139, Abcam), and images were acquired using a Zeiss LSM 980 confocal microscope (Zen Blue, version 3.1)

#### 2.3.3. Isolation of Bone-Marrow-Derived Mesenchymal Stem Cells (BM-MSCs)

For isolation of bone marrow, aspirated bone marrow samples were diluted with complete growth media containing Dulbecco’s Modified Eagle Medium—Low Glucose (DMEM-LG) media (11885084; Gibco, Grand Island, NY, USA) with 10% fetal bovine serum, penicillin–streptomycin (15140163; Gibco, Grand Island, NY, USA), and GlutaMAX supplement (35050079; Gibco, Grand Island, NY, USA), and were seeded in a T-25 culture dish (156367; Nunc, Rochester, NY, USA). The cells were incubated in a humidified atmosphere at 37 °C with 5% CO_2_. Non-adherent cells were removed after 72 h, and, thereafter, the medium was changed every third day until the cell confluency reached 80%. Adherent cells were then subcultured with TrypLE express enzyme and reseeded at 1 × 10^4^ cell/cm^2^.

### 2.4. Characterization of BM-MSCs

#### 2.4.1. Immunophenotyping

At passage 2, the BM-MSCs were assessed for their expression of CD73, CD90, CD105, HLA-DR, DP, DQ, CD45, and CD31. Briefly, the adherent MSCs were harvested by treatment with TrypLE express enzyme, counted, resuspended in 100 µL staining buffer, and then stained with PE-conjugated mouse anti-human CD73 (Cat. No. 344003, 1:100), FITC-conjugated mouse anti-human CD90 (Cat. No. 328107, 1:100), Alexa Fluor 488-conjugated mouse anti-human CD105 (Cat. 323209, 1:100), FITC-conjugated anti-human HLA-DR, DP, DQ (Cat. 361705, 1:100), PerCP mouse anti-human CD45 (Cat. No. 368506, 1:100), and FITC-conjugated mouse anti-human CD31 (Cat. No. 303103, 1:100) at 4 °C for 30 min, and stained cells were detected with LSR Fortessa X-20 (Becton Dickinson, Franklin Lakes, NJ, USA). Cultured MSCs were considered negative for the leukocyte marker CD45, negative for endothelial marker CD31, negative for Immune response marker HLA-DR, and positive for the MSC markers CD90, CD73, and CD105.

#### 2.4.2. Trilineage Differentiation

The trilineage differentiation capacity of BM-MSCs was evaluated using HiAdipoXLTM Adipocyte Differentiation Medium (AL521, Himedia, Thane, MH, India), HiOsteoXLTM Osteocyte Differentiation Medium (AL522, Himedia, Thane, MH, India), and HiChondroXLTM Chondrocyte Differentiation Medium (AL523, Himedia, Thane, MH, India) in 12-well plates as per the manufacturer’s protocol. Briefly, complete media change was performed during the culture period once every 2 to 3 days until they attained 70–80% confluency. The cells were then induced for adipogenesis, osteogenesis, and chondrogenesis, and screened at regular intervals for differentiation. Once the differentiation was observed, cells were fixed with 10% formalin and staining was performed with 0.2% Oil Red O (Sigma-Aldrich, St. Louis, MO, USA) for adipogenic differentiation, 2% Alizarin red S (Sigma-Aldrich, St. Louis, MO, USA) to monitor osteogenic differentiation, and 1% Alcian Blue, pH 2.5 (Sigma, St. Louis, MO, USA) to monitor chondrogenic differentiation. The images were captured using an RTC-7 inverted microscope (Radical, Ambala, HR, India) under a 10× objective. The uninduced cultures expanded in Dulbecco’s Modified Eagle Medium—Low Glucose served as control.

### 2.5. In Vitro Co-Culture Assay

Both direct and indirect co-cultures were performed. For co-cultures, endothelial basal medium (EBM-2; Lonza, Walkersville, MD, USA) without the addition of serum or growth factors (incomplete media) was used.

#### 2.5.1. Indirect Co-Culture of ECFCs and BM-MSCs

For indirect co-culture, transwell chambers with a 0.4 mm pore size membrane (37006, SPL Lifesciences, Pocheon-si, Gyeonggi-do, Republic of Korea) were used. MSCs were seeded in the top hanging insert, while ECFCs were seeded in a separate bottom rat tail collagen type 1-coated 6-well plate with a concentration of 1 × 10^5^ cells in EGM-2. After 4 h, the medium was exchanged for EBM-2, and the hanging insert of BM-MSC was laid over the ECFC well to establish an indirect co-culture dialogue between the two cell types for a further duration of 48 h. After 48 h, the upper transwell was removed, and pure ECFCs from the bottom surface were harvested and maintained at general culture conditions for further experimentation. These ECFCs will be referred to as indirectly primed ECFCs.

#### 2.5.2. Direct Co-Culture of ECFCs and BM-MSCs

The ECFCs were mixed with BM-MSCs at a ratio of 2:1 in a rat tail collagen type 1-coated T-25 flask for an initial period of 4 h in EGM-2, and later changed to EBM-2 for a further duration of 48 h. Monocellular cultures were used as controls. Pure ECFCs were then sorted using CD31 microbeads (130-091-935, Miltenyi Biotec, San Diego, CA, USA) via magnetic-activated cell sorting. The sorted ECFCs were further maintained at general culture conditions, and will be referred to as directly primed ECFCs. To ensure that directly primed ECFCs were free from MSC contamination, harvested cells were further labeled with 1.0 µL of FITC-conjugated mouse anti-human CD31 (Cat. 303,103; BioLegend, San Diego, CA, USA) and 1.0 µL of FITC-conjugated mouse anti-human CD90 (Cat. FAB357P; R&D Systems, Minneapolis, MN, USA) at 4 °C for 30 min, and stained cells were detected using an LSR Fortessa X-20 (Becton Dickinson, Franklin Lakes, NJ, USA). ECFCs were considered to be positive for the expression of CD31, but negative for the expression of CD90. 

### 2.6. Adhesion Assay

The adhesion ability of primed ECFCs was assessed using the method described by Oren M. Tepper et al. [33], with minor modifications. In brief, 1 × 10^5^ primed as well as unprimed ECFCs per well were seeded on a collagen-coated (5 µg/cm^2^ surface area) 24-well plate and incubated for 1 h at 37 °C in 5% CO_2_. Non-adherent cells were washed and remanent adhered cells were fixed with 4% PFA and stained with 0.1% crystal violet for 10 min. The excess stain was rinsed with distilled water and then eluted with 10% acetic acid. The absorbance of the eluted stain was measured at a wavelength of 590 nm using a multimode microplate reader (Spark, Tecan, Seestrasse, Männedorf, Switzerland).

### 2.7. Cellular Proliferation Assay

The metabolic rate and proliferative potential of ECFCs were assessed by performing the 3-(4,5-dimethylthiazol-2-yl)-2,5-diphenyltetrazolium bromide (MTT) colorimetric assay. Briefly, 5 × 10^3^ ECFCs per well were seeded on collagen-coated 96-well plates and incubated at 37 °C with 5% CO_2_. After 48 h, cells were incubated with MTT solution (12 mM) for 4 h at 37 °C. The MTT solution was then aspirated, and formazan crystals were solubilized with 200 µL dimethyl sulfoxide. The absorbance of the dye was measured at a wavelength of 570 nm and recorded using a multimode microplate reader (Spark, Tecan, Seestrasse, Männedorf, Switzerland). The data were normalized so that unprimed cells never exposed to a starved medium were considered to possess 100% activity. The percentage activity was calculated by dividing the absorbance of primed cells and cells in incomplete media by the absorbance of unprimed cells in complete media.

### 2.8. Migration Assay

In vitro migration assay was carried out using 2-well culture inserts as per the manufacturer’s protocol (Ibidi, Germany). Briefly, primed and unprimed ECFCs at 2 × 10^4^ cells per well were seeded in each compartment of a 2-well culture insert and allowed to form a confluent monolayer. The culture inserts were removed creating a uniform cell-free gap of about 500 µm between both monolayers. The wound area filling was captured at 0 h, 3 h, and 6 h using a phase contrast microscope at 10x magnification. The experiment was carried out in duplicate for three independent samples, and the microscopic images were analyzed using ImageJ software (1.53 t; National Institute of Health, Bethesda, MD, USA). The data are expressed as a percentage of migration of the cells between 0–3 h, 3–6 h, and 0–6 h.

### 2.9. In Vitro Tube Formation Assay

In vitro vasculogenesis or angiogenesis was assessed by plating 1 × 10^4^ primed and unprimed ECFCs in 10 µL of growth factor reduced Matrigel matrix (Cat. 354230; Corning) onto µ-slides for angiogenesis (Ibidi, Germany). Cells were imaged using a phase contrast microscope at 4× objective magnification after 12 h of incubation at 37 °C and 5% CO_2_. This assay was performed in duplicate on three independent samples, and the number of nodes, number of junctions, number of segments, the total length of tubules, and the total length of branching capillaries were analyzed using Image J (1.53n; National Institute of Health, Bethesda, MD, USA) software with the angiogenesis analyzer plugin accessed on 9 February 2023 (http://rsb.info.nih.gov/ij/macros/toolsets/Angiogenesis%20Analyzer.txt).

### 2.10. Proteome Profiling

The expression of various proteins related to angiogenesis and other essential processes in cell lysates was assessed using a human angiogenesis proteome profiler array (ARY007, R&D Systems, Minneapolis, MN, USA) as per the manufacturer’s protocol. All experimental groups were maintained in incomplete media to stress the endothelial cells. Unprimed CB-ECFC in complete media served as a healthy baseline control, while unprimed CB-ECFC and BM-MSCs in incomplete media were the unprimed experimental control. Directly and indirectly primed ECFCs are referred to as DP-ECFC (IM) and IDP-ECFC (IM), and considered as experimental groups.

### 2.11. Statistical Analysis 

Analyses were performed using GraphPad Prism v8.3.0 software (GraphPad, La Jolla, CA, USA). Data are presented as mean ± standard deviation (S.D.) of at least three representative experiments. Groups were compared using Student’s *t*-tests and two-way ANOVA. A *p*-value < 0.05 was considered statistically significant.

## 3. Results

### 3.1. Structural and Functional Characterization of CB-ECFC

The CB-ECFCs exhibited a characteristic cobblestoned morphology in culture (Appendix A). They were assessed using flow cytometry based on Medina’s criteria for ECFC characterization [9], and the expression levels of various endothelial markers, including CD31, CD146, and KDR, were 99.63 ± 0.25, 99.93 ± 0.06, and 15.1 ± 1.04, respectively. The expression levels of progenitor markers CD34, CD117, and CD105 were 29.9 ± 11.41, 6.7 ± 0.51, and 99.6 ± 0.26, respectively. The levels of pan-leukocyte marker CD45 and fibroblast marker CD90 were negligible (0.9 ± 0.15 and 2.3 ± 1.15, respectively) (Appendix A).

CB-ECFC positivity for the progenitor marker CD34 and endothelial marker KDR, as well as their uptake of Dil-Acy-LDL and lectin binding, were confirmed through confocal microscopy (Figure 1, Appendix A).

### 3.2. Isolation and Functional Characterization of BM-MSC

The BM-MSCs exhibited a typical fibroblast morphology in culture (Appendix A). They were assessed using flow cytometry based on ISCT guidelines, and the expression level of fibroblast marker CD90 was 97.2 ± 1.94, and progenitor markers CD105 and CD73 were 92.1 ± 9.51 and 96.3 ± 2.33. MSCs were found to be negatively expressing various HLA class II (DR, DP, and DQ), endothelial (CD31), and pan-leukocyte markers (CD45) (1.0 ± 0.12, 0.9 ± 0.15, and 1.2 ± 0.29, respectively) (Appendix A).

BM-MSCs were further assessed for their trilineage potential in adipocytes, osteoblasts, and chondrocytes. After 21 days of culture in the differentiation medium, the verification of differentiation was performed using standard staining methods. The incubation in the adipocyte differentiation medium resulted in the accumulation of lipid droplets in the cytoplasmic region, a characteristic feature of pre-adipocytes, which were stained with Oil Red O (Figure 2A). Osteogenic differentiation-medium-induced BM-MSCs showed deposition of calcium nodules, which were stained orange/yellow with Alizarin Red (Figure 2B), whereas chondrogenic differentiation-medium-induced BM-MSCs showed accumulation of the extracellular matrix, and stained deep blue after staining with Alcian Blue (Figure 2C). Undifferentiated BM-MSCs were kept as a negative control for all the differentiations (Appendix A).

### 3.3. The BM-MSCs Stabilized and Imparted Fibroblast-like Morphology to CB-ECFCs

When cultured in an incomplete medium, CB-ECFCs began to detach from their culture surface after 24 h (Figure 3G), contrary to BM-MSCs, in which no detachments were seen either at 24 h or 48 h (Figure 3H, K). In contrast, CB-ECFCs in direct contact with BM-MSCs formed close associations with BM-MSCs surrounding them (Figure 3I). These associations became stronger after 48 h, and CB-ECFCs appeared to have less detachment when co-cultured with MSCs (Figure 3L) in comparison with CB-ECFCs alone (Figure 3J), suggesting a stabilizing effect of BM-MSC on CB-ECFCs. Pure CB-ECFCs isolated from the direct co-culture showed high expression of endothelial marker CD31 (99.3 ± 0.4) and low expression of fibroblast marker CD90 (2.1 ± 1.7), indicating minimal contamination with BM-MSCs (Appendix A). These sorted CB-ECFCs, henceforth referred to as DP-ECFCs, displayed a more fibroblast-like morphology compared to their unprimed cobblestone-like counterparts. In contrast, CB-ECFCs co-cultured through indirect contact with BM-MSCs in transwell maintained their cobblestone-like morphology.

### 3.4. Priming CB-ECFCs with BM-MSCs Improves the Adhesion Ability of CB-ECFCs

The adhesion ability of CB-ECFCs to the collagen extracellular matrix was significantly reduced when incubated in the incomplete medium (CB-ECFC (IM)). However, priming these impaired CB-ECFCs with BM-MSCs, both directly (DP-ECFC (IM)) and indirectly (IDP-ECFC (IM)), effectively restored their adhesion ability to levels comparable to that of the healthy CB-ECFCs in the complete medium (Figure 4).

### 3.5. Indirectly Primed CB-ECFCs Depicted Better Proliferative Potential Than Directly Primed CB-ECFCs

Direct priming with BM-MSCs (DP-ECFC (IM)) restored the proliferation of unfunctional CB-ECFCs (CB-ECFC (IM)) significantly, but not to the level of CB-ECFCs in the complete medium (CB-ECFC (CM)). Indirect priming of CB-ECFCs (IDP-ECFC (IM)) restored the proliferation to the level of healthy CB-ECFCs in complete medium (CB-ECFC (CM)) (Figure 5). The indirect priming approach also showed significantly better proliferation compared to the directly primed CB-ECFCs.

### 3.6. Priming CB-ECFCs with BM-MSCs Improves Their Migration Ability, with Indirectly Primed ECFCs Migrating Better Than the Directly Primed ECFCs

In the first 3 h of migration, indirectly primed CB-ECFCs (IDP-ECFC (IM)) exhibited significantly better migration ability than directly primed CB-ECFCs (DP-ECFC (IM)) and unprimed dysfunctional CB-ECFCs (CB-ECFC (IM)), whereas directly primed CB-ECFCs showed no significant difference from CB-ECFCs in the incomplete medium. However, in the following 3 h, both directly and indirectly primed CB-ECFCs performed better than the unprimed dysfunctional CB-ECFCs in terms of migration. The migration ability of indirectly primed ECFCs throughout the assay was comparable to that of healthy CB-ECFCs in complete media (CB-ECFC (CM)), and significantly better than both the dysfunctional and directly primed CB-ECFCs, suggesting better migratory potential with indirect priming as compared to direct priming (Figure 6).

### 3.7. Priming CB-ECFCs with BM-MSCs Increases Their Angiogenic Potential

The incomplete media had a significant negative impact on the number of tubes, junctions, segments, total tubule length, and branching length of capillaries formed by CB-ECFCs (CB-ECFC (IM)). Indirect priming (IDP-ECFC (IM)) resulted in a significant improvement in the number of tubes, junctions, and segments, whereas directly primed CB-ECFCs (DP-ECFC (IM)) showed no significant difference from unprimed CB-ECFCs (CB-ECFC (IM)). However, both directly and indirectly primed CB-ECFCs performed significantly better than unprimed CB-ECFCs in terms of the total length of tubules and the total branching length of capillaries in incomplete media (Figure 7).

### 3.8. BM-MSCs Exhibit a Superior Angiogenic Profile Compared to CB-ECFCs

Screening a set of 55 physiologically relevant angiogenesis-related proteins using a human angiogenesis proteome profiler array (ARY007, R&D Systems) showed that 30 proteins were significantly differentially expressed between CB-ECFC (IM) and BM-MSC (IM) (Appendix A). Out of these 30 proteins, 8 growth factors (EGF, FGF-basic, FGF-4, FGF-7, HGF, NRG1ꞵ1, VEGF, VEGF-C) (Figure 8A(a–h)), 7 regulatory proteins (activin-A, coagulation factor III, DPP-IV, IGFBP-2, LAP (TGF-ꞵ1), serpin B5, serpin F1) (Figure 8B(a–g)), 3 cytokines (GM-CSF, leptin, MIP 1α) (Figure 8C(a–c)), and 2 matrix metalloproteinases inhibitory proteins (TIMP-1, TIMP-4) (Figure 8D(a–b)) were highly expressed in BM-MSC (IM), whereas 1 growth factor (PDGF-BB) (Figure 8A(i)), 2 inflammatory cytokines (IL-1ꞵ, IL-8) (Figure 8C(d,e)), 2 extracellular matrix degradation proteins (MMP-8, uPA) (Figure 8D(c,d)), and 5 regulatory proteins involved in angiogenesis, cellular proliferation, and migration (angiopoietin-2, endoglin, endostatin, endothelin-1, thrombospondin-1) (Figure 8B(h–l)) were highly expressed in CB-ECFC (IM) [34]. In brief, BM-MSCs showed a higher level of growth factors, regulatory proteins, and cytokines that promote angiogenesis, while having lower levels of anti-angiogenic proteins, indicating their superior profile in supporting angiogenesis and other related processes. BM-MSCs also expressed low levels of MMPs but high levels of their inhibitors (TIMPs), since they are involved in the production of extracellular matrix (ECM) components that help stabilize vessels. In contrast, CB-ECFCs require the presence of MMPs and uPA, as well as reduced expression of TIMPs, to effectively regulate their migration to a wounded site and support healing. Therefore, a balanced level of these proteins is desirable.

### 3.9. Starvation of CB-ECFCs Initiates Cellular Imbalance 

Out of the 55 analyzed proteins, 13 proteins showed a significant difference between CB-ECFC in a complete medium (CB-ECFC (CM)) and CB-ECFC in an incomplete medium (CB-ECFC(IM)) (Figure 9 and Appendix A).

Out of these 13 proteins, 5 regulatory proteins (angiopoietin-2, endoglin, endostatin, endothelin-1, thrombospondin-1) (Figure 10A(a-e)), 2 extracellular matrix degradation proteins (angiogenin, uPA) (Figure 10B(a,b)), 2 inflammatory cytokines (IL-1ꞵ, IL-8) (Figure 10C(b-c)), 1 molecular marker of endothelial dysfunction (pentraxin 3) (Figure 10D), and 1 growth factor (HB-EGF) (Figure 10E(a)) were significantly higher in CB-ECFC (IM), whereas 1 cytokine (GM-CSF) (Figure 10C(a)) and 1 regulatory protein (LAP/TGF-ꞵ1) (Figure 10A(f)) were significantly reduced in CB-ECFC (IM) in comparison with CB-ECFC at a healthy baseline level. The protein profiles of CB-ECFCs upon starvation of serum indicate overexpression of pro-inflammatory molecules (IL-1β, IL-8, pentraxin 3), anti-angiogenic proteins (angiopoietin-2, endothelin-1, endostatin, thrombospondin-1), and dysregulation of ECM-related proteins (angiogenin, uPA). However, upregulated expression of a few molecules that regulate cell survival and growth (endoglin, HB-EGF) was observed, possibly as an attempt to restore homeostasis.

### 3.10. Direct and Indirect Priming Impart Different Angiogenic Stimuli

On comparing CB-ECFC (IM) with CB-ECFC after priming either by direct (DP-ECFC(IM)) or indirect (IDP-ECFC (IM)) means with BM-MSCs, 21 proteins were found to be differentially expressed. Out of these 21 proteins, 6 proteins, i.e., 2 regulatory proteins (endostatin, thrombospondin-1) (Figure 10A(d,e)), 2 extracellular matrix degradation proteins (angiogenin, uPA) (Figure 10B(a,b)), 1 cytokine (IL-1ꞵ) (Figure 10C(b)), and 1 growth factor (HB-EGF) (Figure 10E(a)), were decreased in both the groups after priming. In total, 16 proteins were significantly altered in both groups, with 3 regulatory proteins (angiopoietin-2, serpin B5, thrombospondin-2) (Figure 10A(a,g,h)) significantly decreased and 1 regulatory protein (endoglin) (Figure 10A(b)) significantly increased in indirectly primed ECFCs, while showing only a beneficial but non-significant trend in directly primed ECFCs. Overall, seven proteins, namely, three growth factors (FGF basic, PD-ECGF, PDGF-BB) (Figure 10E(b,d)), two regulatory proteins (platelet factor 4, serpin F1) (Figure 10A(i–j)), and two matrix metalloproteinases inhibitors (TIMP-1, TIMP-4) (Figure 10B(c,d)) were significantly reduced in directly primed CB-ECFCs (DP-ECFC (IM)), even below the levels in healthy CB-ECFCs (CB-ECFC (CM)), but were unaltered in indirectly primed CB-ECFCs (IDP-ECFC (IM)). Out of the four remnant proteins, one regulatory protein (DPPIV) (Figure 10A(k)) was significantly higher in IDP-ECFC (IM), but lower in DP-ECFC (IM) in comparison with CB-ECFC (IM), and one regulatory protein (endothelin-1) (Figure 10A(c)), one inflammatory cytokine (IL-8) (Figure 10C(c)), and one molecular marker of endothelial dysfunction (pentraxin 3) (Figure 10D) were significantly decreased in IDP-ECFC (IM), but increased in DP-ECFC (IM). These relative protein profiles suggest a more angiogenic profile of IDP-ECFCs in comparison with DP-ECFCs. As a result, we became more interested in investigating the overall distinctions between the two priming methods. In total, 13 proteins were found to be differentially expressed between DP and IDP-ECFCs. Upon analysis, six pro-angiogenic proteins, namely, DPPIV, endoglin, FGF basic, PD-ECGF, uPA, and VEGF-C (Figure 11A(a–f)), were increased in IDP-ECFCs in comparison with DP-ECFC (IM), three anti-angiogenic proteins, namely, endothelin-1, pentraxin 3, and thrombospondin-2 (Figure 11B(a–c)), were significantly decreased in IDP-ECFCs in comparison with DP-ECFCs, and two inflammatory proteins, namely, IL-8 and MCP-1 (Figure 11C(a,b)), were significantly decreased in IDP-ECFCs, but were higher than CB-ECFC (IM) in DP-ECFCs, depicting excessive inflammation in DP-ECFCs. Although the expression of serpin F1 and TIMP-1 (Figure 11D(A,B)) increased in IDP-ECFCs, their actual role in angiogenesis is still debatable.

## 4. Discussion

In the present study, we comprehensively assessed the functional parameters of ECFCs primed with MSCs, and found that priming with BM-MSCs, either direct (DP-ECFC (IM)) or indirect (IDP-ECFC (IM)) priming improved proliferative potential, migration ability, and angiogenic potential of ECFCs as compared to ECFCs in starved media (absence of growth factors and serum) (CB-ECFC (IM)). Further, we observed that indirectly primed ECFCs performed better than directly primed ECFCs in all functional aspects without compromising the self-renewal potential. Immunomagnetic separation to sort the directly primed ECFCs may be an additional stressor to these cells, thereby affecting their functionalities.

Shafiee et al. (2016) observed an improvement in ECFC survival and proliferation after priming them with MSCs in transwell, but did not observe substantial changes in terms of tube formation potential [32]. Popescu et al. (2021), in a similar study, reported the enhancement of vasculogenic potential of ECFCs upon co-culturing with MSC-conditioned medium. [35].

We also reported distinct angiogenic profile of ECFC and MSCs individually. Differential expression of angiogenic and migratory proteins was also observed in the primed ECFCs. Co-culturing ECFCs with MSCs has been shown to enhance the expression of proteins involved in cellular adhesion, chemotaxis, endogenous cell repair, and vasculogenesis [35] in the secretome of the co-cultured cells, which comprises two fractions: the soluble fraction, which contains soluble proteins or cytokines, and the extracellular vesicle (EV) fraction, which changes with microenvironment [36,37]. Unlike these studies, we looked at the angiogenic profile of the cellular fraction. To optimize the use of ECFCs in cell therapy, it is crucial to understand the cellular changes that occur during their interaction with MSCs. Our results show that BM-MSCs had a more pro-angiogenic profile, with increased expression of growth factors and decreased expression of anti-angiogenic proteins. On the other hand, CB-ECFCs exhibited a more migratory profile, with increased expression of ECM-degradation-related proteins and decreased levels of their inhibitors. In ECFCs cultured in starved media, a significant increase in the expression of angiopoietin-2, endothelin-1, endostatin, and thrombospondin-1 was observed (angiopoietin-2 inhibits ECFC migration, sprouting, and survival, and promotes vessel destabilization, especially in the absence of VEGF [38,39,40]; endothelin-1 indicates the onset of uncontrolled vasoconstriction in endothelial cells [41]; endostatin inhibits the binding of VEGF-C to VEGFR-2 [42]; and thrombospondin-1 is an antagonist of the activity of VEGF [43]) and a decrease in the LAP/TGF-β1 ligand. However, the upregulated expression of endoglin, a receptor of TGF-β1, was seen, possibly indicating homeostasis and endothelial cell activation [44,45]. Plasmin formation proteins, such as uPA and angiogenin [46], and some growth factors, such as HB-EGF, were also found to be significantly increased. A significant increase in inflammatory cytokines, such as IL-1β and IL-8, an indicator of local vascular inflammation, and pentraxin 3, a molecular marker of endothelial cell dysfunction [47,48], were also observed, supporting our experimental findings of dysfunctional CB-ECFCs upon serum deprivation. Comparison of the profile of indirectly primed CB-ECFCs with CB-ECFCs from starved media suggested a significant restoration of various regulatory proteins (angiopoietin-2, endothelin-1, endostatin, thrombospondin-1), with a further decrease in the expression of various anti-angiogenic proteins (serpin B5 and thrombospondin-2), ECM degradation proteins (angiogenin and uPA), cytokines (IL-1β, IL-8, and pentraxin 3), and HB-EGF, suggesting a stabilizing effect of BM-MSC on CB-ECFCs in the indirect co-culture. Endoglin and DPPIV, which promote cell interactions with the ECM, showed an increase. Whereas when directly primed, ECFCs were compared with ECFC (IM), and only a few regulatory proteins (endostatin, and thrombospondin-1), ECM-degradation-related proteins (angiogenin, uPA), cytokine (IL-1β), and growth factor (HB-EGF) showed a significant reversal. The profile of directly primed ECFCs depicted an increased expression of endothelin-1 and IL-8, suggesting excessive inflammation, which is also evident from an overexpression of pentraxin 3, and, overall, indicating vascular inflammation. In addition, in directly primed ECFCs, the expression of certain regulatory proteins (DPPIV), MMP inhibitors (TIMP-1 and TIMP-4), and growth factors (FGF-basic, PD-ECGF, PDGF-BB) essential for the proper functioning of cells decreased to a level below that of even the ECFC (CM), suggesting an undesirable effect that needs to be explored further. Comparison of expression profiles of directly and indirectly primed ECFCs showed increased expression of six pro-angiogenesis proteins (DPPIV, endoglin, FGF basic, PD-ECGF, uPA, and VEGF-C) and a decrease in three anti-angiogenic proteins (endothelin-1, pentraxin 3, thrombospondin-2) in indirectly primed ECFCs. An increased expression of pro-inflammatory cytokines (IL-8 and MCP-1) in directly primed ECFCs suggests vascular inflammation, which possibly makes them less suitable for cellular therapies.

In conclusion, our findings indicate that indirectly primed ECFCs are better candidates for cellular therapies (Figure 12). However, thorough in vivo experimentations are needed to verify these findings before they can be used in translational applications. Moreover, to better understand the modulations occurring during MSC and ECFC interactions, transcriptome analysis studies, such as RNA sequencing, are recommended. Such studies will help identify the molecular mediators and understand the actual pathways responsible for imparting these effects on CB-ECFCs, as well as help predict better outcomes of cell therapy and accelerate their escalation towards therapeutics.

## Figures and Tables

**Figure 1 biomedicines-11-01372-f001:**
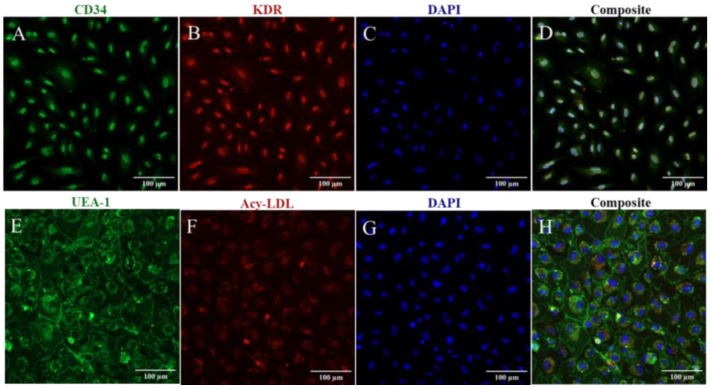
The confocal microscopy images depicting the expression of progenitor and endothelial markers in cord-blood-derived endothelial colony-forming cells and uptake of AcLDL and Ulex europaeus agglutinin (UEA-1) binding: (**A**) expression of progenitor marker CD34 (green); (**B**) expression of endothelial marker KDR (red); (**C**) nuclear stain with DAPI (blue); (**D**) composite image; (**E**) binding to Ulex europaeus agglutinin (UEA-1); (**F**) uptake of Dil-Acy LDL; (**G**) nuclear stain with DAPI (blue); (**H**) composite image. All the images are at 20× magnification. Scale bar, 100 µm.

**Figure 2 biomedicines-11-01372-f002:**
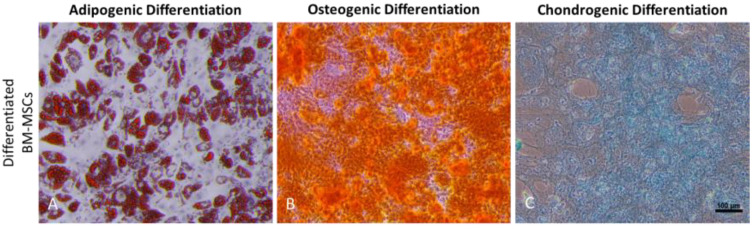
Trilineage differentiation potential of human bone-marrow-derived MSCs as described by the International Society for Cell & Gene Therapy (**A**) depicting adipocytes differentiated from BM-MSCs; (**B**) depicting osteoblasts differentiated from BM-MSCs; and (**C**) depicting chondrocytes differentiated from BM-MSCs. All the images are at 10× magnification. Scale bar, 100 µm.

**Figure 3 biomedicines-11-01372-f003:**
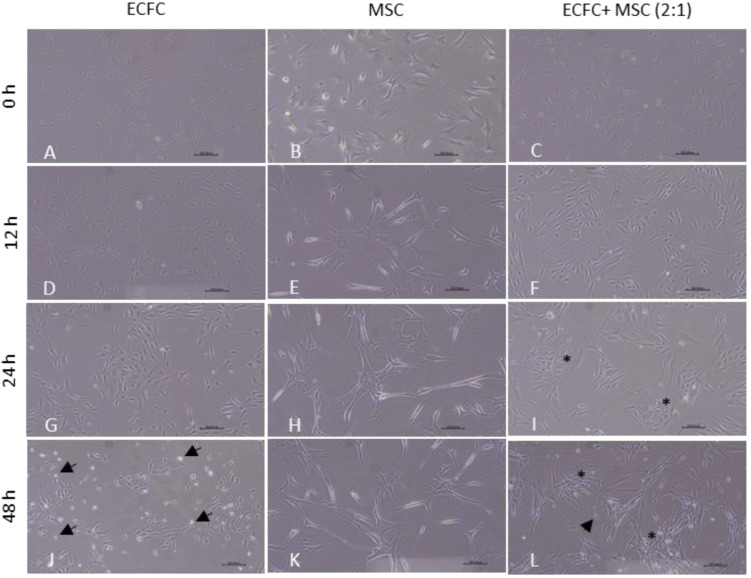
Pocketing of CB-ECFCs by BM-MSCs during direct co-culture. Representative images of ECFCs, BM- MSCs, and ECFCs:BM-MSCs (2:1) at (**A**–**C**) 0 h, (**D**–**F**) 12 h, (**G**–**I**) 24 h, and (**J**–**L**) 48 h respectively in serum- and growth-factor-deprived conditions; asterisks (*) indicate pocketing of ECFCs surrounded by MSCs, and arrows indicate detachment of ECFCs that are not in contact with MSCs (10× magnification; scale bar, 300 µm).

**Figure 4 biomedicines-11-01372-f004:**
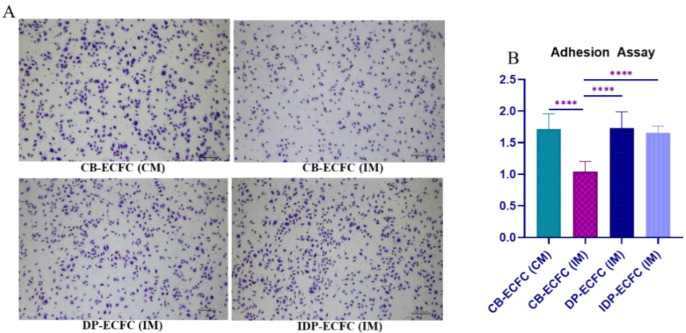
(**A**) Brightfield microscopic image of adhered ECFCs stained with crystal violet to determine cellular adhesion (4× magnification; scale bar, 200 µm); (**B**) bar graph depicting mean absorbance at 590 nm, bar represents mean ± SD, *n* = 3, **** *p* ≤ 0.0001. Significance was assessed by a two-way ANOVA.

**Figure 5 biomedicines-11-01372-f005:**
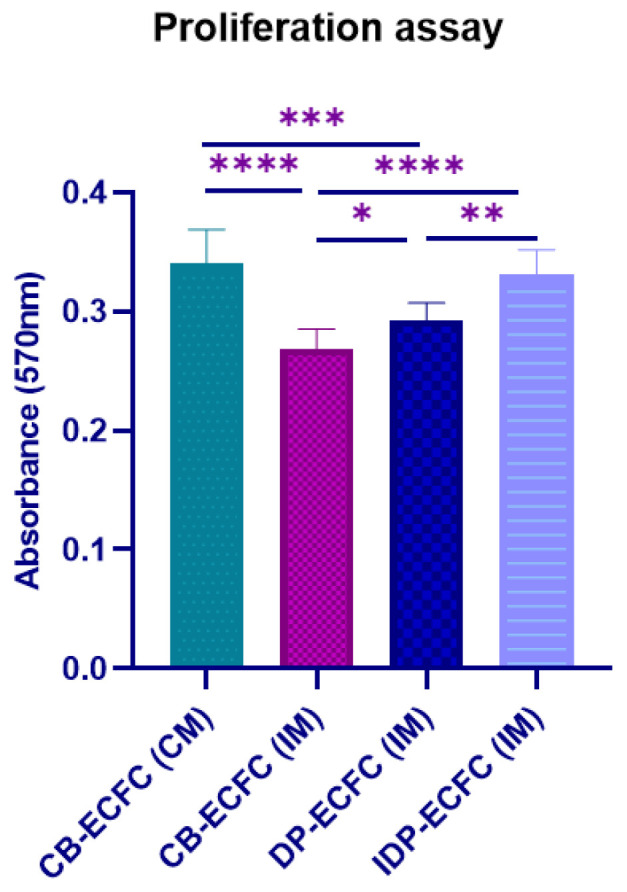
Bar graph depicting absorbance at 570 nm; bar represents mean ± SD, *n* = 3, * *p* ≤ 0.05, ** *p* ≤ 0.01, *** *p* ≤ 0.001, **** *p* ≤ 0.0001. Significance was assessed by a two-way ANOVA.

**Figure 6 biomedicines-11-01372-f006:**
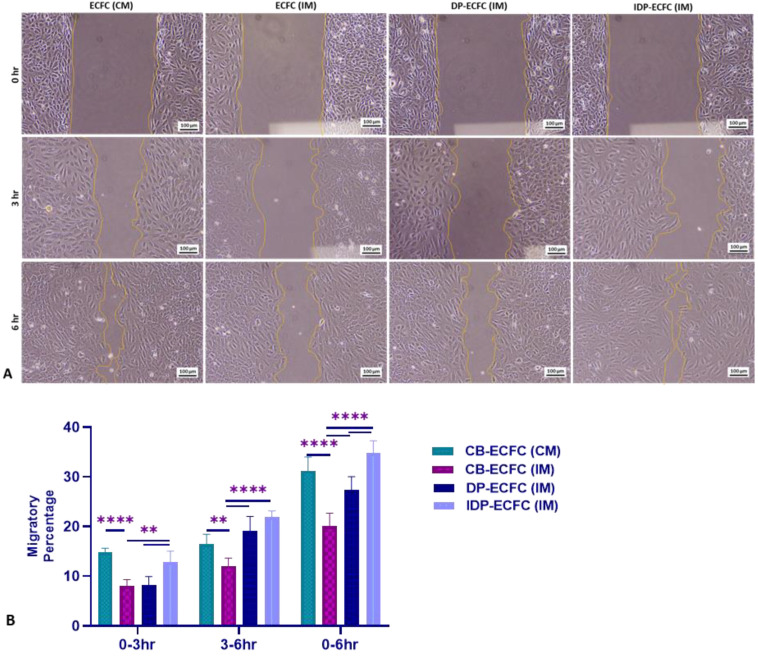
(**A**) Representative phase contrast images showing migration potential of ECFCs (10× magnification; scale bar, 100 µm); (**B**) bar graph depicting the mean migratory percentage during the specified time interval; bar represents mean ± SD, *n* = 3, ** *p* ≤ 0.01, **** *p* ≤ 0.0001. Significance was assessed by a two-way ANOVA.

**Figure 7 biomedicines-11-01372-f007:**
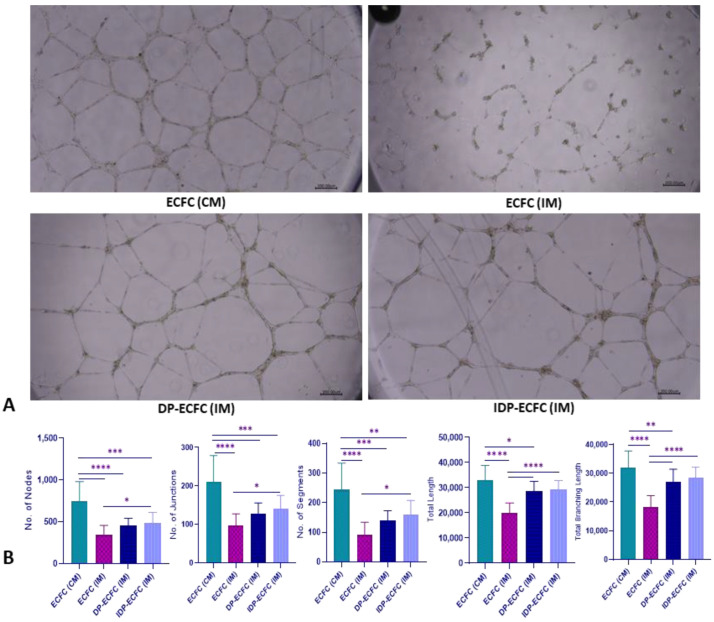
(**A**) Phase contrast images of tubules formed by ECFCs (4× magnification; scale bar, 200 µm); (**B**) bar graph depicting the mean number of nodes ± SD, mean number of junctions ± SD, mean number of segments ± SD, mean total length of tubules ± SD, and the mean branching length of capillaries ± SD, *n* = 3, * *p* ≤ 0.05, ** *p* ≤ 0.01, *** *p* ≤ 0.001, **** *p* ≤ 0.0001. Significance was assessed by a two-way ANOVA.

**Figure 8 biomedicines-11-01372-f008:**
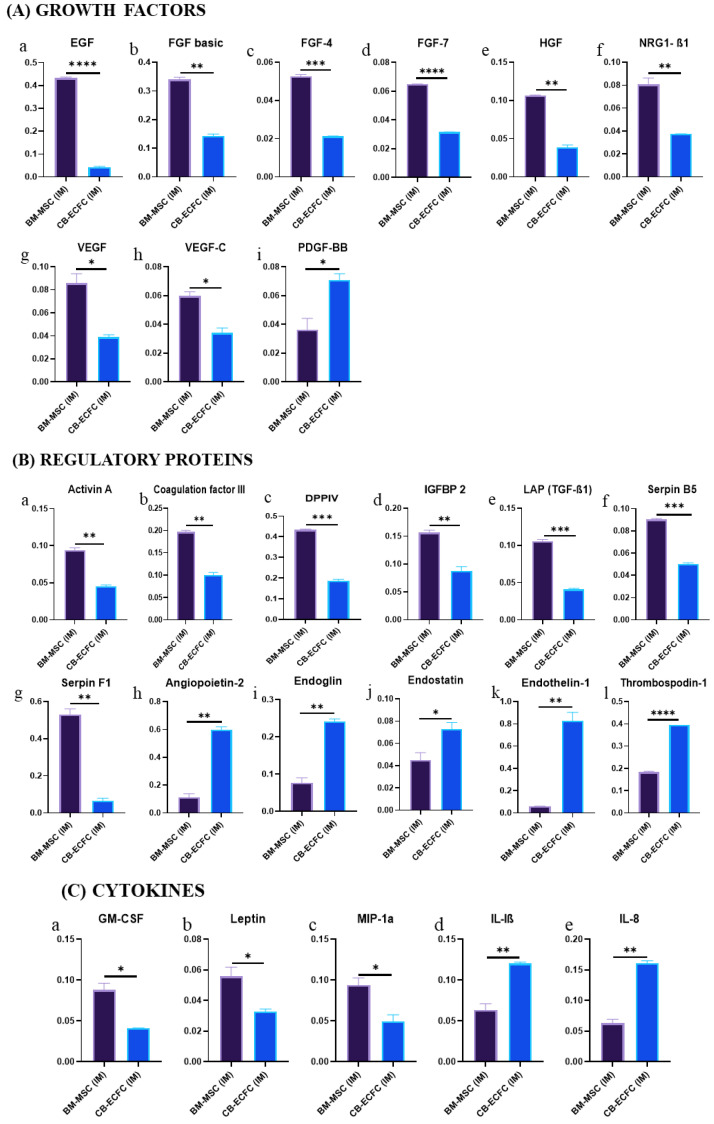
Bar graphs depicting relative expression of various (**A**(**a**–**i**)) growth factors, (**B**(**a**–**l**)) regulatory proteins, (**C**(**a**–**e**)) cytokines, (**D**(**a**–**d**)) MMP inhibitors, and ECM-degradation-related proteins, in BM-MSCs and CB-ECFCs; bar represents mean ± SD, *n* = 2, * *p* ≤ 0.05, ** *p* ≤ 0.01, *** *p* ≤ 0.001, **** *p* ≤ 0.0001. Significance was assessed by Student’s *t*-test.

**Figure 9 biomedicines-11-01372-f009:**
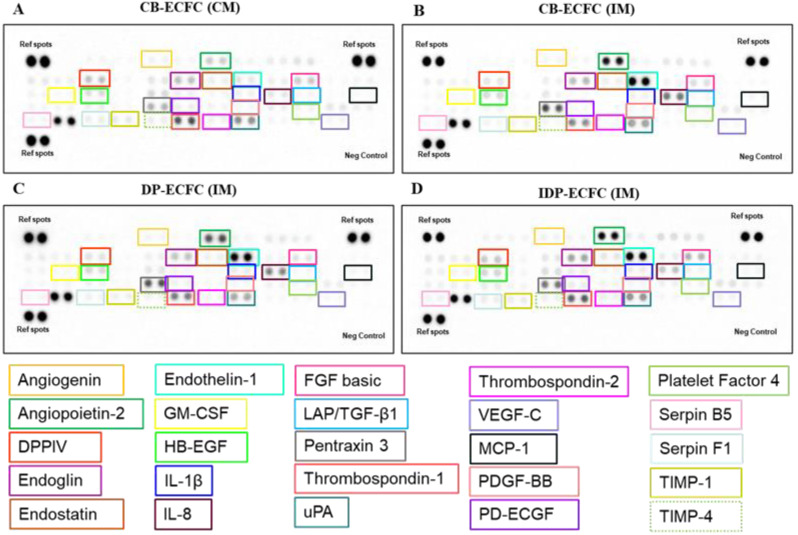
Representative images of angiogenesis array membranes of various groups depicting relative expression of various proteins: (**A**) healthy CB-ECFC (CM); (**B**) impaired CB-ECFC (IM); (**C**) directly primed ECFC (DP-ECFC (IM)); (**D**) indirectly primed ECFC (IDP-ECFC (IM)).

**Figure 10 biomedicines-11-01372-f010:**
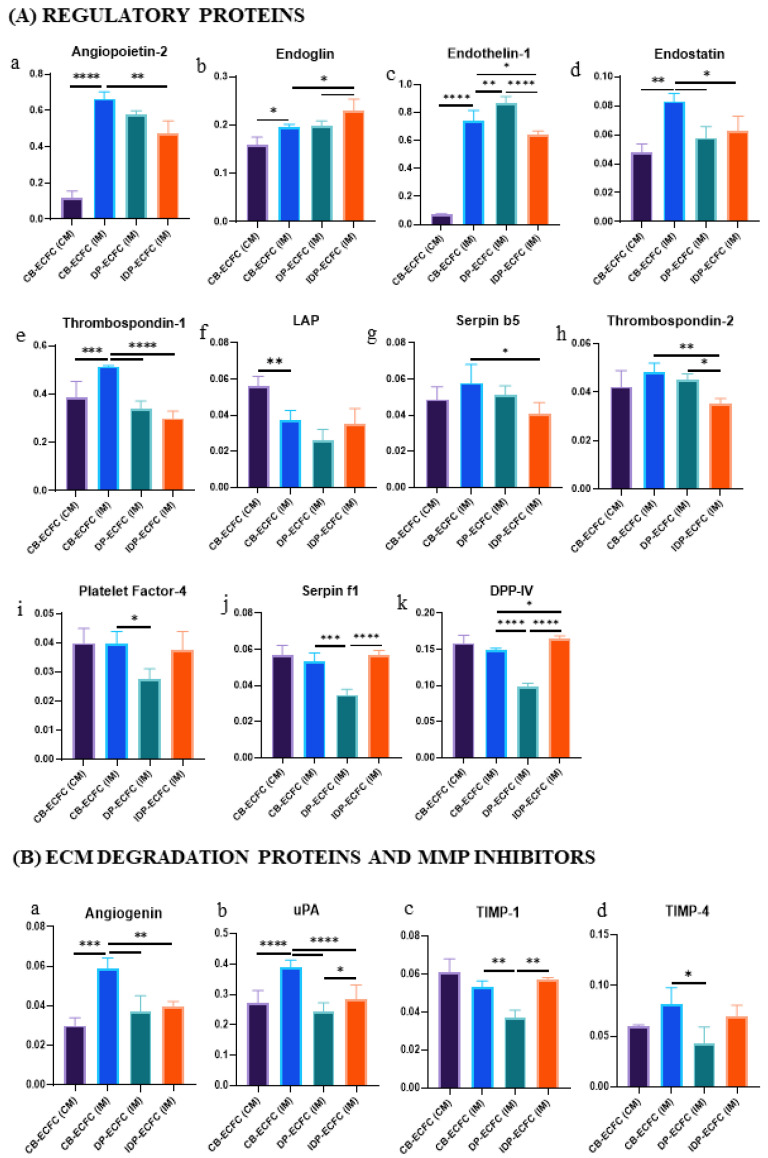
Bar graphs depicting relative expression of various (**A**(**a**–**k**))regulatory proteins, (**B**(**a**–**d**)) ECM degradation proteins and MMP inhibitors, (**C**(**a**–**c**)) cytokines, (**D**) endothelial dysfunction, and (**E**(**a**–**d**)) growth factors; bar represents mean ± SD, *n* = 3, * *p* ≤ 0.05, ** *p* ≤ 0.01, *** *p* ≤ 0.001, **** *p* ≤ 0.0001. Significance was assessed by a two-way ANOVA.

**Figure 11 biomedicines-11-01372-f011:**
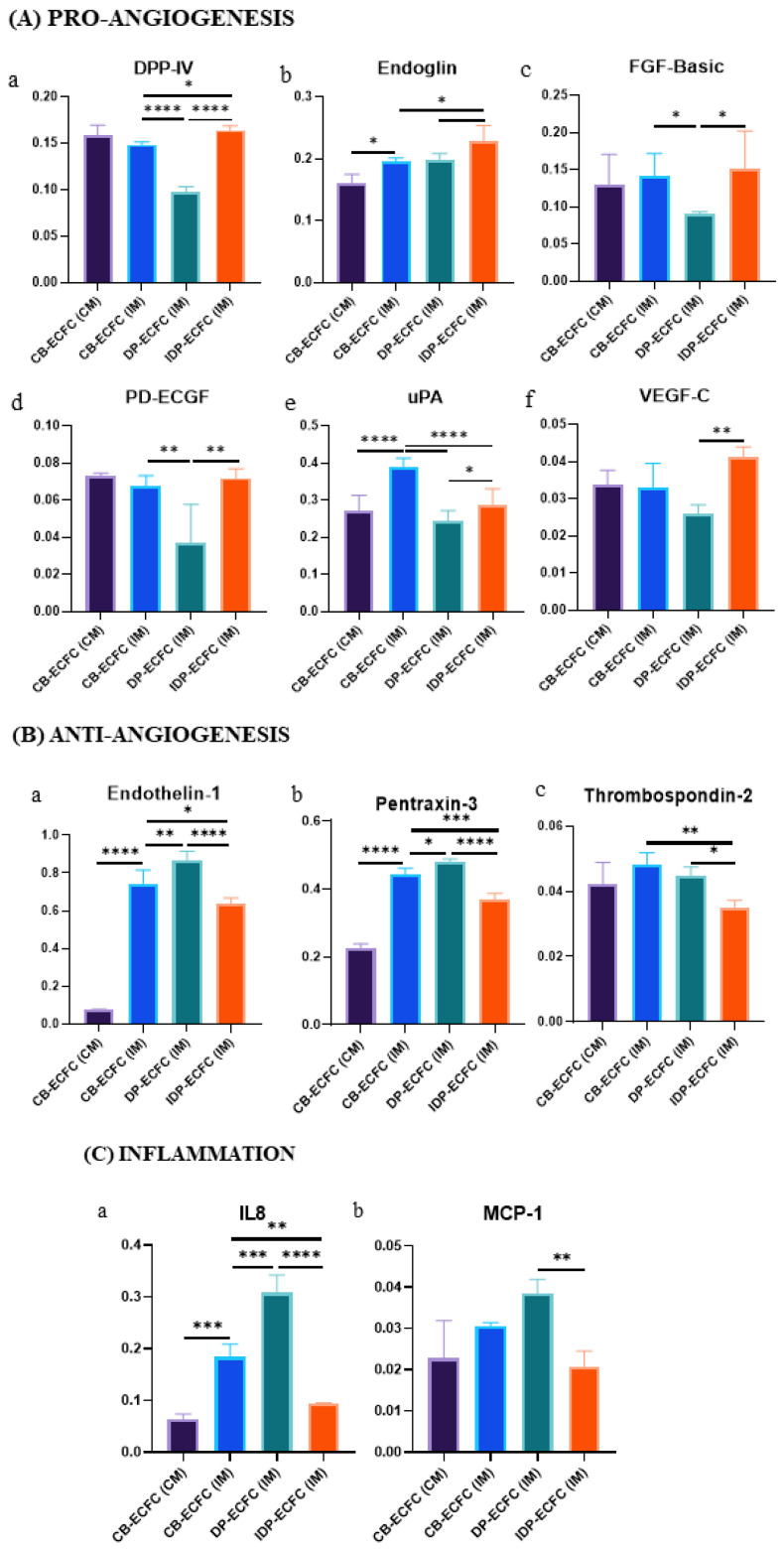
Bar graphs depicting relative expression of various (**A**(**a**–**f**)) pro-angiogenic proteins, (**B**(**a**–**c**)) anti-angiogenic proteins, (**C**(**a**,**b**)) inflammation-related proteins, and (**D**(**A**,**B**)) other proteins; bar represents mean ± SD, *n* = 3, **p* ≤ 0.05, ***p* ≤ 0.01, ****p* ≤ 0.001, *****p* ≤ 0.0001. Significance was assessed by a two-way ANOVA.

**Figure 12 biomedicines-11-01372-f012:**
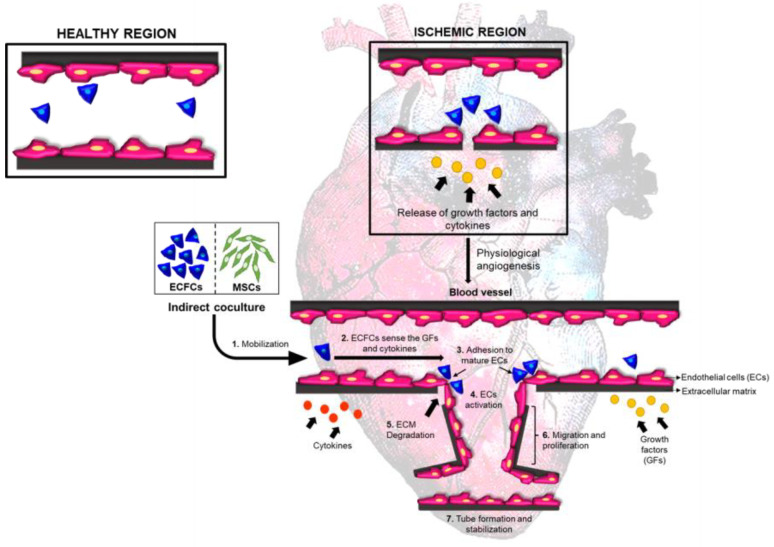
Schematic diagram summarizing various events involved in the revascularization process through recruitment of ECFCs.

## Data Availability

All relevant data appear in the paper. The datasets used and/or analyzed in the current study are available from the corresponding author upon reasonable request.

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
