# Peer review of "Augmenting the Angiogenic Profile and Functionality of Cord Blood Endothelial Colony-Forming Cells by Indirect Priming with Bone-Marrow-Derived Mesenchymal Stromal Cells"

_biomedicines, 2023, doi:10.3390/biomedicines11051372_

Round 1
Reviewer 1 Report
Augmenting Angiogenic Profile and Functionality of Cord Blood-Endothelial Colony Forming Cells by Indirect Priming with Bone Marrow-Derived Mesenchymal Stromal Cells by Ashutosh Bansal et al.
In the abstract sent by the editor, Lakshmy Ramakrishnan was the second author. However, in the submitted and downloaded manuscript is listed as a last author. This is a big change in authorship. What was the reason for that? The Abstract sent out to reviewers cannot contain such misleading information. The potential reviewer is sometimes searching, based on the last author’s name, for previous publications from the same group.
This is an interesting manuscript adding more knowledge to the CB-ECFC/MSC field. The authors isolated and characterized both cell types and differentiated MSCs into three lineages. They explored the impact of MSCs on ECFCs when co-cultured in direct and indirect contact, mimicking contact-mediated and paracrine-mediated effects. They also assessed the effects of MSCs on ECFCs cultured in incomplete media. Several functional assays were performed for proliferation, adhesion, migration and capillary tube formation. In addition, proteins involved in vasculogenesis were assessed by a protein array panel. ECFCs (in incomplete media) with indirect MSCs contact appeared to show an enhanced vasculogenic potential based on their proteomic signature.
The results overall show that the stress of culturing ECFCs in incomplete media is alleviated by indirect MSCs contact. The authors should emphasize more that they used incomplete media (withdrawing media components) as a method of endothelial cell stress. It can have only limited relevance to any ischemic condition occurring in vivo.
The reviewer could not access the supplementary figures so the complete evaluation of the manuscript was not possible!
Comments and suggestions:
Abstract
The abstract is hard to read, is confusing at times with some inaccuracies. Please rewrite it in a simple and clear manner.
Page 1, line 19: “Recent reports claim the improvement of ECFC vasculogenic potential…”
Page 1, line 19-20: ECFC, MSCs: please write full name at first mention.
Page 1, line 27: “Both directly and indirectly primed ECFCs significantly restored the engraftment and….” There are no data presented related to ECFCs engraftment.
Line 28: “…whereas indirectly primed ECFCs showed better metabolism…” The MTT assay used and the graph presented provide data on cellular proliferation (based on SDH enzyme activity). Metabolism is a general term and includes a vast array of cellular processes.
Methods
2.3.1 and 2.4.1. Immunophenotyping and Line 206:
What was the cell suspension volume for staining with the antibodies listed? Alternatively, give the dilution used for the antibodies.
Line 239: the 6h time point is missing
Line 256: “All experimental groups were maintained in incomplete media to simulate the in-vivo condition during a CVD.” This cannot simulate an in-vivo condition, rather can serve as a stress condition for the cells.
Results
Line 291: “…were not expressing various MHC class…”
Figure 3: the two different cell types cannot be recognized based only on shape. On 3B, all cells appear to be elongated and changed compared to 3A. Detached cells cannot be identified as which cell type detached. Some detached cells appear to be associated with cell clusters. Staining with at least one marker for each cell type would help to identify different cells and their location.
Line 337 (Figure 4 legend): “…590 nm.” In the Methods is given 570 nm.
Direct priming (with mixed cells) may look less efficient when compared to direct priming because the cells had to be subjected to an additional step of immunomagnetic separation. This may affect their function.
Figures 4B and 5, 6, 7, 8, 10, 11: please give “Mean+/- St Dev or SEM, n=…” in the figure legends
Page 10, line 383: “Screening a set of 55 physiologically relevant angiogenesis-related proteins showed…” How did you choose these proteins? Please give citations.
Page 10, lines 394-402: Please provide citations for these statements.
Page 14, line 471 (Figure 10. Figure legend): “(IV) proliferation and migration related protein” On the graph it is marked as “Endothelial dysfunction” (Pentraxin-3).
Supplementary Materials: 10.6084/m9.figshare.22187131 via this link cannot be found. Please compress the files and upload them or combine them with the manuscript.
Discussion
Page 14,
Line 484: “…in starved media…” Please indicate specifically what was withdrawn from the media.
line 489: “Popescu et al (2021) in a similar study reported enhancement of functionality of ECFCs upon coculture with MSCs.” Please be more specific. What functions were improved/changed?
Lines 487, 489: please cite the papers with the numeric citations also […].
Page 15,
Line 505: “…and thrombospondin-1 was observed.”
Line 543: “Such studies will unveil the molecular mediators and decode the actual pathways responsible for imparting these effects on CB-ECFCs,…” This is an overstatement, since RNA seq can only suggest/hypothesize some molecular mediators and pathways. It will never be able to unveil and decode the real, complex mechanisms.
Author Response
In the abstract sent by the editor, Lakshmy Ramakrishnan was the second author. However, in the submitted and downloaded manuscript is listed as a last author. This is a big change in authorship. What was the reason for that? The Abstract sent out to reviewers cannot contain such misleading information. The potential reviewer is sometimes searching, based on the last author’s name, for previous publications from the same group.
We thank the reviewer for pointing out this important information. While submitting the manuscript, Dr. Saumitra Dey Choudhury was mistakenly marked as the corresponding author. Whereas, Prof. Lakshmy Ramakrishnan remains the corresponding author and last author of this manuscript.
This is an interesting manuscript adding more knowledge to the CB-ECFC/MSC field. The authors isolated and characterized both cell types and differentiated MSCs into three lineages. They explored the impact of MSCs on ECFCs when co-cultured in direct and indirect contact, mimicking contact-mediated and paracrine-mediated effects. They also assessed the effects of MSCs on ECFCs cultured in incomplete media. Several functional assays were performed for proliferation, adhesion, migration and capillary tube formation. In addition, proteins involved in vasculogenesis were assessed by a protein array panel. ECFCs (in incomplete media) with indirect MSCs contact appeared to show an enhanced vasculogenic potential based on their proteomic signature.
The results overall show that the stress of culturing ECFCs in incomplete media is alleviated by indirect MSCs contact. The authors should emphasize more that they used incomplete media (withdrawing media components) as a method of endothelial cell stress. It can have only limited relevance to any ischemic condition occurring in vivo.
We agree with the reviewer’s comments and accordingly have changed the statement and have now emphasized that the endothelial cell stress was created by withdrawing media components (page 6, line 256).
The reviewer could not access the supplementary figures so the complete evaluation of the manuscript was not possible!
We had uploaded a link for supplementary figures but now we have appended them to the manuscript.
Comments and suggestions:
Abstract
The abstract is hard to read, is confusing at times with some inaccuracies. Please rewrite it in a simple and clear manner.
The abstract is now made simple and clear for ease of understanding.
Page 1, line 19: “Recent reports claim the improvement of ECFC vasculogenic potential…”
Page 1, line 19-20: ECFC, MSCs: please write full name at first mention.
As suggested, we have expanded ECFC and MSC (Line 19-21, Page 1)
Page 1, line 27: “Both directly and indirectly primed ECFCs significantly restored the engraftment and….” There are no data presented related to ECFCs engraftment.
As pointed out by the reviewer engraftment may not be the appropriate term to use, accordingly we have changed it to adhesion (Line 27, Page 1; Line 327, Page 8).
Line 28: “…whereas indirectly primed ECFCs showed better metabolism…” The MTT assay used and the graph presented provide data on cellular proliferation (based on SDH enzyme activity). Metabolism is a general term and includes a vast array of cellular processes.
We thank reviewer for the comment and have removed the term metabolism and have restricted to proliferation (Line 28, Page 1; Line 220, Page 5; Line 346, Page 8).
Methods
2.3.1 and 2.4.1. Immunophenotyping and Line 206:
What was the cell suspension volume for staining with the antibodies listed? Alternatively, give the dilution used for the antibodies.
We have included the details in the revised manuscript (Line 110-117, Page 3; Line 158- 164, Page 4).
Line 239: the 6h time point is missing
We have now included the 6hr time point in the revised manuscript (Line 238, Page 5).
Line 256: “All experimental groups were maintained in incomplete media to simulate the in-vivo condition during a CVD.” This cannot simulate an in-vivo condition, rather can serve as a stress condition for the cells.
We agree with the reviewer’s comments and have accordingly changed the statement (Line 256, Page 6).
Results
Line 291: “…were not expressing various MHC class…”
We have changed MHC to HLA in the revised manuscript (Line 290-291, Page 7).
Figure 3: the two different cell types cannot be recognized based only on shape. On 3B, all cells appear to be elongated and changed compared to 3A. Detached cells cannot be identified as which cell type detached. Some detached cells appear to be associated with cell clusters. Staining with at least one marker for each cell type would help to identify different cells and their location.
As shown in figure 8 (supplementary) the ECFCs alone when grown for 48 hours showed detachment (supplementary figure 8 D) where as MSC alone did not show any rounding or detachment of cells (supplementary figure 8-H). Based on this, it can be inferred that only ECFCs detach after 48 hours, however ECFCs in contact with MSC seems to remain adherent (supplementary figure 8-L). When figure 3 and supplementary figure 8 are viewed together this fact becomes clear.
Line 337 (Figure 4 legend): “…590 nm.” In the Methods is given 570 nm.
We thank reviewer for pointing this out. We have now changed the wavelength to 590 in methodology (Line 218, Page 5).
Direct priming (with mixed cells) may look less efficient when compared to direct priming because the cells had to be subjected to an additional step of immunomagnetic separation. This may affect their function.
We agree with the reviewer and therefore the results of our study also conclude that indirect priming is more efficient. We have included this line in the discussion (Line 489-491, Page 14).
Figures 4B and 5, 6, 7, 8, 10, 11: please give “Mean+/- St Dev or SEM, n=…” in the figure legends
We have given mean and SD and in the revised manuscript have now even “n” also (Line 337, Page 8; Line 348, Page 9; Line 365, Page 9, Line 380, Page 10; Line 410, Page 12; Line 473, Page 14).
Page 10, line 383: “Screening a set of 55 physiologically relevant angiogenesis-related proteins showed…” How did you choose these proteins? Please give citations.
This is a commercial angiogenic proteome profiler kit with pre-defined set of angiogenic/antiangiogenic proteins (Line 383-384, Page 10).
Page 10, lines 394-402: Please provide citations for these statements.
We have provided the citation for the segregation of proteins on the basis of their roles. (Line 395, Page 10)
Page 14, line 471 (Figure 10. Figure legend): “(IV) proliferation and migration related protein” On the graph it is marked as “Endothelial dysfunction” (Pentraxin-3).
We thank the reviewer for pointing this out. We have now changed “proliferation and migration-related protein” to “Endothelial Dysfunction” in the legend (Line 473, Page 14).
Supplementary Materials: 10.6084/m9.figshare.22187131 via this link cannot be found. Please compress the files and upload them or combine them with the manuscript.
We have now combined them with the manuscript.
Discussion
Page 14,
Line 484: “…in starved media…” Please indicate specifically what was withdrawn from the media.
We have included the necessary details as pointed out by the reviewer (Line 486- 487, Page 14).
line 489: “Popescu et al (2021) in a similar study reported enhancement of functionality of ECFCs upon coculture with MSCs.” Please be more specific. What functions were improved/changed?
We have specified the enhanced functionality of ECFCs upon co-culturing with MSC’s Conditioned medium. (Line 494-495, Page 15).
Lines 487, 489: please cite the papers with the numeric citations also […].
The numeric citations of the paper are now included. (Line 495-496, Page 15)
Page 15,
Line 505: “…and thrombospondin-1 was observed.”
We thank the reviewer for pointing this out. We have now added “was observed”. (Line 511, Page 15).
Line 543: “Such studies will unveil the molecular mediators and decode the actual pathways responsible for imparting these effects on CB-ECFCs,…” This is an overstatement, since RNA seq can only suggest/hypothesize some molecular mediators and pathways. It will never be able to unveil and decode the real, complex mechanisms.
We agree with the reviewer and have now modified the statement as pointed out by the reviewer. (Line 557-558, Page 16).
Reviewer 2 Report
The manuscript ”Augmenting Angiogenic Profile and Functionality of Cord Blood-Endothelial Colony Forming Cells by Indirect Priming with Bone Marrow-Derived Mesenchymal Stromal Cells” brings scientific evidence for improving cellular therapy in cardiovascular pathology. Angiogenesis is a biological process that can be manipulated under particular circumstances, as similar articles reveal. Experimental methodology is adequately used to argue the use of Endothelial Colony Forming cells and Mesenchymal Stromal cells for pro-angiogenesis induction.
1. The minor revision is oriented toward reducing the overuse of abbreviations and making the article more reader-friendly.
2. A more underlined connection with the clinical relevance of the article is needed.
3. Schematic figures will help understand the scientific arguments supporting angiogenesis promoted by co-cultured cells.
Author Response
The manuscript ”Augmenting Angiogenic Profile and Functionality of Cord Blood-Endothelial Colony Forming Cells by Indirect Priming with Bone Marrow-Derived Mesenchymal Stromal Cells” brings scientific evidence for improving cellular therapy in cardiovascular pathology. Angiogenesis is a biological process that can be manipulated under particular circumstances, as similar articles reveal. Experimental methodology is adequately used to argue the use of Endothelial Colony Forming cells and Mesenchymal Stromal cells for pro-angiogenesis induction.
- The minor revision is oriented toward reducing the overuse of abbreviations and making the article more reader-friendly.
We thank the reviewer for pointing this out. We have tried to reduce the abbreviations wherever possible and have kept them to a minimum.
- A more underlined connection with the clinical relevance of the article is needed.
This manuscript aims to augment the functionality of ECFCs after priming them indirectly with MSCs. We presently have limited the clinical linkage of the article as the in-vitro experiments can help identify possible approaches to improvise the presently less promising cell therapies. Whereas, thorough in-vivo experimentation will be required to clearly state the efficiency of these approaches in translational aspects. (Line 553-554, Page 16)
- Schematic figures will help understand the scientific arguments supporting angiogenesis promoted by co-cultured cells.
We have included a schematic figure (Figure 12.) summarizing the various events involved in the revascularization process through the recruitment of ECFCs (Line 548-550, Page 16).
Round 2
Reviewer 1 Report
The authors addressed all comments and concerns. One minor point: please replace Figure 3 with Supplementary Figure 8 to make this argument more convincing (e.g MSCs alone do not appear to detach after 48h as opposed to CB-ECFCs. CB-ECFCs appear to have less detachment when cocultured with MSCs).
Author Response
The authors addressed all comments and concerns. One minor point: please replace Figure 3 with Supplementary Figure 8 to make this argument more convincing (e.g. MSCs alone do not appear to detach after 48h as opposed to CB-ECFCs. CB-ECFCs appear to have less detachment when cocultured with MSCs).
We thank the reviewer for his/her time and efforts in thoroughly reviewing the manuscript. As suggested, we have replaced Figure 3 with Figure 8 from supplementary and made the necessary changes.